# Maternal Diet May Modulate Breast Milk Microbiota—A Case Study in a Group of Colombian Women

**DOI:** 10.3390/microorganisms11071812

**Published:** 2023-07-14

**Authors:** Diana C. Londoño-Sierra, Victoria Mesa, Nathalia Correa Guzmán, Laura Bolívar Parra, Olga I. Montoya-Campuzano, Sandra L. Restrepo-Mesa

**Affiliations:** 1Food and Human Nutrition Research Group, School of Nutrition and Dietetics, Antioquia University, Medellín 050010, Colombia; dcarolina.londono@udea.edu.co (D.C.L.-S.); victoria.mesa-schein@u-paris.fr (V.M.); nathalia.correag@udea.edu.co (N.C.G.); 2Physiopathologie et Pharmacotoxicologie Placentaire Humaine Microbiote Pré & Postnatal (3PHM), INSERM, UMR-S 1139, Université Paris Cité, 75006 Paris, France; 3Probiotics and Bioprospecting Research Group, Faculty of Sciences, National University of Colombia, Medellín 050034, Colombia; lbolivarp@unal.edu.co (L.B.P.); oimontoy@unal.edu.co (O.I.M.-C.)

**Keywords:** breast milk microbiota, maternal nutrition, nutritional status

## Abstract

There is increasing evidence that the diet and nutritional status of women during pregnancy and lactation can modulate the microbiota of their milk and, therefore, the microbiota of the infant. An observational, descriptive, and cross-sectional study was carried out in a group of lactating women. Dietary intake during gestation and the first trimester of lactation was evaluated, and the microbiota was analyzed by 16S ribosomal RNA (rRNA) sequencing using the Illumina platform. Globally, *Streptococcus* spp. (32%), *Staphylococcus* spp. (17.3%), *Corynebacterium* spp. (5.1%) and *Veillonella* spp. (3.1%) were the predominant bacterial genera. The consumption of simple carbohydrates in gestation (rho = 0.55, *p* ≤ 0.01) and lactation (rho = 0.50, *p* ≤ 0.01) were positively correlated with *Enterobacter* spp. In lactation, a negative correlation was observed between the intake of simple carbohydrates and the genus *Bifidobacterium* spp. (rho = −0.51 *p* ≤ 0.01); furthermore, a positive correlation was identified between the intake of folic acid and *Akkermansia* spp. (rho = 0.47, *p* ≤ 0.01). Amplicon sequence variants (ASVs) associated with the delivery mode, employment relationship, the baby’s gender, birth weight, the Body Mass Index (BMI) of the breastfeeding woman, and gestational weight gain were recovered as covariates in a linear mixed model. The results of this research showed that the maternal nutritional status and diet of women during gestation and lactation could modulate the microbiota of breast milk.

## 1. Introduction

The role of the human microbiota in healthy growth and development during the first years of life has been related to its beneficial metabolic and structural functions, the regulation of immunity and systemic inflammation, as well as its influence on the somatotrophic axis, which has regulated the production of growth factors such as the insulin-like growth factor 1 (IGF1) and growth hormone, as well as energy and nutritional metabolism [1,2]. Moreover, the first interaction of microorganisms with humans is fundamental for immunological, metabolic, and systemic adaptation; the first 1000 days of life are considered a decisive immunological window for health in the later stages of life [3].

Due to its enormous metabolic capacity, the microbiota is considered essential for life, with great influence on health and disease. The population composition of the microbiota is particular and exhibits its own characteristics in each individual; therefore, it varies according to genetics, the mode of birth, the type of feeding in the early years, habitual diet, the use of probiotics, exposure to antibiotics, interactions with the environment, among other factors [4]. It is estimated that about 25–30% of the infant microbiota has its origin in breast milk [2], in which a central bacteriome composed of nine genera, *Staphylococcus* spp., *Streptococcus* spp., *Serratia* spp., *Pseudomonas* spp., *Corynebacterium* spp., *Ralstonia* spp., *Propionibacterium* spp., *Sphingomonas* spp., and *Bradyrhizobium* spp. has been identified [5]. A recent study conducted in Colombia on breast milk samples from women donors identified that the milk microbiota contained commensal microorganisms, including, among them, lactic acid bacteria with probiotic potential using culture methods [6]; however, no studies have been developed in a national context to identify how a woman’s diet modulates the microbiota of her milk.

It has been documented that breast milk microbiota is modulated by several factors such as diet and weight [5]. Regarding maternal feeding, the consumption of carbohydrates, fiber and vegetable proteins could influence the abundance of *Staphylococcus* spp., *Bifidobacterium* spp., and *Lactobacillus* spp. [7], and associations have also been found between the intake of polyunsaturated fatty acids and the genus *Bifidobacterium* spp. [8]. Micronutrients such as calcium and vitamin B2 have been positively associated with *Veillonella* spp. abundance [9] and vitamin C intake with a higher abundance of *Staphylococcus* spp. [8].

In relation to the Body Mass Index (BMI) and gestational weight gain, changes in breast milk microbiota have also been identified. The BMI of lactating women has been negatively related to the genus *Bacteroides* spp. [9] and gestational weight gain has shown a positive relationship with alpha diversity in breast milk microbiota [10]. It has been described that excess maternal weight could generate changes in the milk’s metabolome and in the microbiota: mechanisms that could be involved in the risk of infant obesity [11,12,13].

The connection between nutritional status, a woman’s diet during gestation and lactation, and the microbiota of breast milk highlights the need to go deeper into this subject in order to identify aspects that, from a dietary and nutritional point of view, could be modified in favor of the microbiota, which can contribute to favoring the health of the mother-child binomial in the short, medium and long term. The complexity of breast milk and the factors that contribute to its composition are essential aspects that must be taken into account in order to expand our knowledge and design strategies targeting breast milk microbiota to influence maternal and infant health. The objective of this study was to analyze the effects of food and nutritional status during gestation and the first trimester of lactation on the microbiota of breast milk in a group of healthy lactating women in Colombia.

## 2. Materials and Methods

An observational, descriptive, and cross-sectional study was carried out on a group of breastfeeding women who had prenatal care in two health institutions in eastern Antioquia, Colombia.

### 2.1. Subjects

The sample consisted of 30 women in their first trimester of lactation who were selected at convenience and who had a prenatal medical history in the referenced institutions (Figure 1).

The criteria for the selection of mothers were the following: breastfeeding women between 18 and 39 years of age who had a singleton pregnancy, without diseases or complications during gestation and postpartum (anemia, diabetes, hypertensive disorders, among others), with adequate BMI or overweight, at least three prenatal controls, with food security at home according to the Latin American and Caribbean Food Security Scale (ELCSA) [14], who had a full-term newborn, and who exclusively breastfed their child. Women with obesity (BMI > 30 kg/m^2^) or thinness (<18.5 kg/m^2^) and women who, during gestation, lactation, and up to 30 days prior to data collection, chronically consumed medications or other substances such as antibiotics, antidepressants, laxatives, corticosteroids, cigarettes, alcohol, proton pump inhibitors, and probiotics were excluded.

The participants were identified at the health institution; then, their medical history was reviewed, and if they met the inclusion criteria, they were contacted to sign the informed consent form and to schedule two visits for data collection.

#### 2.1.1. Collection of Anthropometric Data

The women’s weight and height were measured using digital scales and a portable measuring rod, and with these data the BMI (BMI = kg/m^2^) was calculated, which according to the World Health Organization (WHO), was classified as adequate (≥18.5 to <25 kg/m^2^) or overweight (≥25 kg/m^2^ to <30 kg/m^2^). To collect anthropometric information in the gestational stage, the participants’ clinical history was asked for in terms of their pregestational maternal weight and the weight reported in the prenatal controls. Total weight gain during gestation was determined by the difference between the pregestational weight and the last weight reported in their clinical history. Adequate gestational weight gain was considered based on the adjusted allowed ranges for the Pregestational Body Mass Index (PBMI) according to the references of the Institute of Medicine of the United States (IOM): underweight 12.5 and 18 kg; normal weight 11.5–16 kg; overweight 7 and 11.5 kg [15]; weight gain was classified as excessive when it exceeded these ranges and insufficient when it did not reach the minimum expected. Newborn weight and length data were taken; birth weight was classified as insufficient at 2500–2999 g and adequate between 3000 and 3999 g [16].

#### 2.1.2. Evaluation of Food Consumption

Two 24 h recalls (R24h) were applied to each participant, using the adjusted multi-step technique on non-consecutive days, and were distributed during the week: a procedure necessary to adjust for intra- and inter-individual variability [17]. To determine the amount of food intake, a set of models, geometric figures, and an album of photographs with life-size utensils were used, which are validated for Colombia [18].

To identify the aspects relating to nutrition during pregnancy, a quantitative frequency of consumption was applied, which was composed of 81 foods selected from the consumption reported in pregnant women in the Food and Nutrition Profile of the Department of Antioquia 2019 [19]. This included foods such as legumes, fruits, and vegetables for their contribution of soluble and insoluble fiber [20]; sources of saturated, unsaturated, and polyunsaturated fatty acids [21] such as lard, vegetable margarine, butter, industrialized sauces, vegetable oil, olive, canola, nuts, and seeds; fermented foods such as yogurt [22]; animal proteins such as beef, pork, chicken, fish, and eggs; industrialized meats; and foods high in simple and ultra-processed carbohydrates [23].

To test the estimation of nutrients from the quantitative frequency of foods, a concordance test was performed between the designed frequency and the R24h in a group of seven pregnant women belonging to the prenatal control program of one of the reference institutions to whom both instruments were applied for subsequent analysis. To estimate these differences, the Wilcoxon signed-rank test, the Hodges-Lehmann test and 95% confidence intervals (95% CI) were applied; to evaluate this correlation, the Biserial Rank Correlation Coefficient (95% CI) was applied and to evaluate the concordance, the Concordance Correlation Coefficient was also used (95% CI) (Appendix A). The data collected for this test were not included in the study.

#### 2.1.3. Food Consumption Analysis

For nutrient analysis, the R24h was processed in the Dietary Intake Evaluation software (EVINDI v5) of the School of Nutrition and Dietetics of the University of Antioquia [24], and the database obtained was migrated into the Personal Computer Software for Intake Distribution Estimation (PC-SIDE v1.0) [25], which is available at the Department of Statistics at Iowa State University, Ames IA (United States). For each nutrient, the Estimated Average Requirement (EAR) established in the Recommendations for Energy and Nutrient Intake for the Colombian population was used as a cut-off point. For macronutrients, the Acceptable Macronutrient Distribution Range (%AMDR) was recommended in the national guidelines [26].

The prevalence of the risk of deficiency was accompanied by summary measures such as the minimum, maximum, percentiles, mean, and standard deviation, which were adjusted in PC-SIDE v1.0 [25]. The contribution of nutrients of interest was obtained: calories, proteins, total fat, saturated, monounsaturated, polyunsaturated, cholesterol, total and simple carbohydrates, dietary fiber, zinc, calcium, iron, magnesium, vitamins: B1, B2, B3, B5, B6, B9, B12, A, and C. Data processing and analysis were performed in SPSS v25, EVINDI v5.0 and PC-SIDE v1.0 software.

### 2.2. Collection of Breast Milk Samples

The samples were collected between 8 and 10 a.m., manually, and from the breast opposite to the last suckling of the newborn or from the breast opposite to the one from which the baby was suckling. The nipple and the surrounding area were cleaned with sterile gauze and 0.5% chlorhexidine. Between 15 and 20 mL of milk were collected, discarding the first drops, and the milk was deposited in sterile tubes free of RNAses and DNAses (Corning Incorporated, Corning, NY, USA). After the collection process, the milk samples were transported in a cooler with dry ice to the Food and Human Nutrition Research Laboratory, University of Antioquia, Colombia (transport time less than 1 h), where they were stored at −80 °C for subsequent DNA extraction.

### 2.3. Extraction, Quantification and Sequencing of Barcoded Amplicons on the Illumina MiSeq Platform

The total genomic DNA extraction was performed from 6 to 10 mL of breast milk using the GeneJET Genomic DNA Purification kit (Thermo Scientific) at the molecular biology laboratory of the Faculty of Agricultural Sciences, University of Antioquia, Colombia. The extracted DNA was quantified using the 260/280 optical density ratio by UV absorbance methods (NanoDrop, Thermo Scientific, Wilmington, NC, USA). Hypervariable regions V3-V4 of the 16S ribosomal ribonucleic acid (rRNA) gene were amplified using 1 μL of DNA (25 ng on average). A Polymerase Chain Reaction (PCR)was performed in 27 cycles within the following reaction conditions: 95 °C for 20 s, 55 °C for 20 s, and 72 °C for 20 s. The primers used were Bakt_341F: CCTACGGGGGNGGCWGCAG and Bakt_805R: GACTACHVGGGGGTATCTAATCC, and each sample was assigned a unique 6-base pair (bp) barcode. Barcoded PCR products were purified from triplicate reactions with an agarose gel band purification kit (Illustra GFX PCR dna and gel Band Purification Kit, GE Healthcare, UK). Equimolar concentrations of PCR amplicons were quantified by fluorometric methods (Qubit 3.0—Thermo Fisher Scientific, Waltham, MA, USA). Purified amplicons were pooled in equimolar amounts (~50 ng per sample) for library preparation. Sequencing was performed using the Illumina MiSeq paired-end platform (2 × 300 base pairs) with 100,000 reads for each library (Macrogen, Korea).

### 2.4. Analysis of Microbiota Data

These sequences were demultiplexed, thus removing the primer sequences and associated barcodes. The bioinformatics analysis of sequences was performed in QIIME2 (Quantitative Insights into Microbial Ecology) v2019.7 software [27]. The DADA2 method [28] was used to detect and correct sequencing noise, remove chimeric sequences, and cluster sequences into amplicon sequencing variants (ASVs). To classify these sequences according to their taxonomic information, the qiime feature-classifier plugin was employed using the Vsearch alignment method [29] with the SILVA v138 database [30] at a 99% sequence identity. Subsequently, the BIOM (Biological Observation Matrix Data) table (frequency table of each ASV with its taxonomic assignment), the phylogenetic tree, and the metadata of samples with the information of variables under study were imported for analysis in the RStudio v1.1.453 software [31].

### 2.5. Statistical Analysis

For the descriptive analysis of sociodemographic, gestational, anthropometric, and food consumption characteristics, absolute and relative distributions and summary indicators such as the arithmetic mean and standard deviation were used. To compare the diversity and richness of the bacterial community, alpha diversity was analyzed using four indices: the Chao1 index, which estimates the richness of taxa in a community [32], the Shannon index, which allows an evaluation of the heterogeneity of a community based on the number of species present and their relative abundance [33]; Simpson’s inverse index, also known as the dominance index, which allows measurements of the richness of organisms [34] and the PD index, which describes diversity based on phylogenetic distances [35]. Comparisons between these groups were performed using non-parametric tests: the Wilcoxon rank test or Kruskal–Wallis. For beta diversity, principal coordinate analysis (PCoA) was performed to identify a clustering pattern of microbial compositions as a function of the variables of interest using permutation-based methods (PERMANOVA, permuted multivariate analysis of variance, using the Adonis2 library) for weighted and unweighted UniFrac distances [36]. Diversity analyses were performed using the Phyloseq [37] and Microbiome [38] packages of the Rstudio v4.1.2 software [31].

Correlations between the most abundant and literature-reported bacterial genera and nutrient intake values were explored using Spearman’s correlation coefficient, which was visualized by a heatmap in the RStudio program using the Corrplot package [39]. Through a mixed linear model (LMM), the association of the most abundant ASVs transformed logarithmically with the individual variables registered in the metadata and were evaluated using the RStudio’s lme4 and nlme packages [40,41]. Later, with the ASVs that showed a significant association (*p* < 0.05), the model was adjusted to control the effects of other variables so that the resulting variation explained was independent of other variables and not subject to confusion by the correlated variables; each variable was the fixed factor and the others entered as covariates.

The study followed the ethical considerations established in the Declaration of Helsinki and was validated by the Bioethics Committee of the Faculty of Dentistry of the University of Antioquia: concept No. 66-2020, Act No. 10 of 2020. Informed consent was obtained from all participants.

### 2.6. Data Availability Statement

Sequence reads were deposited in the European Nucleotide Archive (ENA) via the project number PRJEB59523.

## 3. Results

### 3.1. Sociodemographic, Gestational and Anthropometric Characteristics

The average age of the lactating women was 25 ± 6 years; 73% had the presence of a partner, 37% had finished higher education, 57% lived in a rural area, and 53% belonged to the subsidized health regime. Of the total number of participating women, 60% did not plan their pregnancy, 53% had between 1 and 2 children, 80% attended more than six prenatal check-ups, and 43% were first-time mothers.

In relation to the anthropometric characteristics, the average weight of the lactating women was 60.8 ± 7.9 kg, 60% presented a normal BMI %, and the majority presented a height ≥ 1.55 m (63%). In relation to gestation, most of the women started with a BMI in adequacy (70%), and at the end, the average weight gained was 12.2 ± 3.6 kg; 43% presented inadequate weight gain due to deficiencies, and 13% presented with an inadequate gain due to excess. More than half of them had a vaginal delivery (77%); the newborns presented an average of 3299 ± 275 g and 49.8 ± 1.7 cm at birth, were breastfed in the first hour of life 87%, and were male 63% (Table 1).

### 3.2. Nutrient Intake

The mean adjusted energy intake was 2185 calories (Standard Deviation (SD) = 399), the prevalence of risk of deficiency in usual energy intake was 43% (SD = 0.11), and the risk of excess was 16% (SD = 0.12). Regarding the consumption of macronutrients, the prevalence of the risk of deficiency in the usual intake of proteins was 99% (SD = 0.03), the consumption above the reference value of total fat (>35% AMDR) was 3% (SD = 0.12) and of total carbohydrates (>65% AMDR) was 1% (SD = 0.06). Regarding the consumption of nutrients of interest, the mean adjusted intake of cholesterol was 493 mg (SD = 145), the consumption above the reference value (>10% ADMR) of saturated fat was 86% (SD = 0.18), and for simple carbohydrates was 72% (SD = 0.10); 97% of the women did not reach the recommended fiber intake (Table 2).

In relation to micronutrient intake, the highest prevalence in the risk of deficiency of usual intake was presented for folic acid at 87% (SD = 0.10), vitamin C at 65% (SD = 0.10), vitamin B6 at 60% (SD = 0.11), thiamine at 57% (SD = 0.17), magnesium at 53% (SD = 0.10), zinc at 52% (SD = 0.09), niacin at 52% (SD = 0.10) and vitamin A at 50% (SD = 0.12). By contrast, the lowest prevalence in the risk of deficiency of usual intake was presented in iron intake 9.1% (SD = 0.10), vitamin B12 7.3% (SD = 0.10), riboflavin 0.6% (SD = 0.04) and the prevalence of a low risk of deficiency in pantothenic acid intake was 18.8% (SD = 0.10) as presented in Table 2. Regarding the consumption of supplements during gestation, all participants consumed iron and folic acid supplements 97%, which was not the case with calcium supplementation, where only 37% reported daily consumption; however, it should be noted that due to the consumption of dairy products, the risk of calcium deficiency was low.

In the survey of the mother’s food intake during gestation, it was found that the median energy intake was 2.553 calories (median absolute deviation, MAD = 597.5), protein 94 g (MAD = 24.5), total fat 71 g (MAD = 17.1), carbohydrates 379 g (MAD = 97), simple carbohydrates 76 g (MAD = 36.1) and dietary fiber 24 g (MAD = 8.1). Regarding the intake of nutrients of interest, the median intake of saturated fat was 28 g (MAD = 6.95), and simple carbohydrates was 76 g (MAD = 36.1); for micronutrient intake, the highest median intakes during gestation were for vitamin A intake 1394.5 ER (MAD = 443.5) and folic acid 1461.5 mg (MAD = 257) Table 3.

### 3.3. Bioinformatic Analysis of the Milk Microbiota

After filtering, merging, and checking for chimera sequences of the 16S RNA gene, in the V3-V4 region, from the 30 samples collected, a total of 2,940,975 sequences were obtained. The sequencing depth of the data processed by the DADA2 method [28] ranged from 56,744 to 126,815 sequences per sample (Appendix A). Rarefaction curves showed the number of taxa (richness or alpha diversity) as a function of the sample size or the number of reads. Most samples reached the plateau effect, indicating that the diversity of all sequences obtained was sampled (Appendix A).

### 3.4. Characterization of the Milk Microbiota

In the characterization of the microbiota of breast milk samples from lactating women in Colombia, 25 bacterial phyla were found, the most abundant being: Firmicutes (69.5%), Actinobacteria (10%), Proteobacteria (9.6%), and Bacteroidetes (7.6%). The bacterial phyla identified are listed in Appendix A.

Regarding the bacterial genera, 644 were detected, with the most abundant being *Streptococcus* spp. (32%), *Staphylococcus* spp. (17.3%), *Corynebacterium* spp. (5.1%) and *Veillonella* spp. (3.1%). Ten genera were identified with a relative abundance between 1.2% and 2.6%, *Bacteroides* spp. (2.6%), *Lactobacillus* spp. (2.4%), *Bacillus* spp. (1.9%), *Rothia* spp. (1.8%), *Methylobacterium* spp. (1.6%), *Clostridum sensu stricto* 1 spp. (1.5%), *Pseudomonas* spp. (1.4%), *Fusobacterium* spp. (1.4%), *Gemella* spp. (1.3%) and *Prevotella* spp. (1.3%) (Figure 2).

#### Differences in Alpha and Beta Diversity in Relation to the Variables under Study

A trend toward higher bacterial richness and diversity was observed in the milk samples from women with a C-section delivery (Chao 1 *p* = 0.022; PD *p* = 0.03) (Figure 3A) and those who had a direct employment relationship or through a family member (Chao1 *p* = 0.029; PD *p* ≤ 0.01) (Figure 3B). No differences were found according to the area of residence, maternal age, level of schooling, the number of previous pregnancies, or the sex of the infant; however, it was identified that breast milk microbiota from women with male infants showed a tendency to have a greater richness and diversity (Chao1 *p* = 0.74; PD *p* = 0.06) (Figure 3C). In newborns with insufficient birth weight, there was also a tendency for a lower richness and diversity in the microbiota compared to those with an adequate birth weight (Chao1 *p* = 0.99; PD *p* = 0.84) (Figure 3D).

According to the nutritional status by anthropometric indicators, a tendency to have a higher alpha diversity through the Shannon index and InvSimpson index was observed in women with an adequate BMI without statistical differences (Shannon *p* = 0.22; InvSimpson *p* = 0.18) (Figure 4A); a lower richness and diversity was also observed in those women who had excessive gestational weight gain (Chao1 *p* = 0.51; PD *p* = 0.96) (Figure 4B).

The principal coordinate analysis (PCoA) was performed to identify a clustering pattern of microbial composition based on the weighted UniFrac distances, where the distance represented the difference between microbial communities, taking into account phylogenetic distances and unweighted UniFrac where they are not taken into account. Based on the weighted UniFrac distances, the beta diversity of breast milk microbiota presented differences according to gestational weight gain (PERMANOVA *p* = 0.033). The pairwise beta-diversity comparisons between groups showed significant overall differences across the groups, with higher distance dissimilarities among groups belonging to inadequacies by deficiencies and excess, which was observed when compared to the reference adequate. The variables under analysis explained the variability of the microbiota in 63.5% for the weighted UniFrac distances and 11.8% for the unweighted UniFrac distances (Figure 5).

### 3.5. Relationship of Nutrient Intake during Lactation with the Microbiota of Breast Milk

The intake of macro and micronutrients during lactation showed positive correlations with the microbiota, which meant that the higher the intake of a nutrient, the higher the abundance of a bacterial genus. In relation to macronutrient intake, a positive correlation was identified between the consumption of simple carbohydrates and *Enterobacter* spp. (rho = 0.50, *p* ≤ 0.01); on the other hand, the intake of total fat (rho = 0.39, *p* = 0.03), saturated fat (rho = 0.38, *p* = 0.03), and monounsaturated fat (rho = 0.42, *p* = 0.02) showed a positive correlation with the genus *Eubacterium* spp. (Figure 6).

Regarding micronutrient intake, a positive correlation was identified between folic acid intake and *Akkermansia* spp. (rho = 0.47, *p* ≤ 0.01); between B complex vitamins such as B1 (rho = 0.45, *p* = 0.01), B2 (rho = 0.51, *p* ≤ 0.01), B3 (rho = 0.46, *p* ≤ 0.01) and the genus *Gemella* spp.; as well as vitamin A intake and the genera *Bifidobacterium* spp. (rho = 0.36, *p* = 0.047), *Corynebacterium* spp. (rho = 0.43, *p* = 0.01) and *Ruminococcus* UCG.009 spp. (rho = 0.49, *p* ≤ 0.01) (Figure 6).

There were negative or inverse correlations where the higher the consumption of a nutrient, the lower the abundance of a bacterial genus or in the opposite direction. Among these, several macro and micronutrients presented negative correlations with the genus *Aerococcus* spp.; among this was the consumption of the total protein (rho = −0.45, *p* = 0.01), total carbohydrates (rho = −0.47, *p* ≤ 0.01), cholesterol (rho = −0.49, *p* ≤ 0.01), dietary fiber (rho = −0.61, *p* ≤ 0. 01), vitamin A (rho = −0.46, *p* = 0.01), vitamin C (rho = −0.58, *p* ≤ 0.01), folic acid (rho = −0.51, *p* ≤ 0.01), pantothenic acid (rho = −0.49, *p* ≤ 0.01), magnesium (rho = −0.58, *p* ≤ 0.01) (Figure 6).

Saturated fat intake showed a negative correlation with several bacterial genera: *Corynebacterium* 1 spp. (rho = −0.49, *p* ≤ 0.01), *Cutibacterium* spp. (rho = −0.36, *p* = 0.047),

*Escherichia-Shigella* spp. (rho = −0.51, *p* ≤ 0.01) and Lachnospiraceae NK4A136 (rho = −0.36, *p* = 0.049); a negative correlation was also observed between the intake of simple carbohydrates and the genus *Bifidobacterium* spp. (rho = −0.51 *p* ≤ 0.01), as well as dietary fiber and *Enterobacter* spp. (rho = −0.36, *p* = 0.047) (Figure 6).

### 3.6. Relationship of Nutrient Intake during Gestation with Breast Milk Microbiota

Positive correlations were identified between the intake of macronutrients such as simple carbohydrates (rho = 0.55, *p* ≤ 0.01), total carbohydrates (rho = 0.39, *p* = 0.02), saturated fat (rho = 0.39, *p* = 0.03) and the total protein (rho = 0.3, *p* = 0.04) with the genus *Enterobacter* spp. Saturated fat intake was also positively correlated with the genus *Halomonas* spp. (rho = 0.41, *p* = 0.02). Regarding micronutrient intake, positive correlations were observed between zinc intake and *Pseudomonas* spp. (rho = 0.38, *p* = 0.03), vitamin C and *Rothia* spp. (rho = 0.44, *p* = 0.01) (Figure 6).

Protein intake (rho = −0.46, *p* = 0.01) and saturated fat (rho = −0.52, *p* ≤ 0.01) were negatively correlated with *Bifidobacterium* spp.; on the other hand, dietary fiber intake was negatively correlated with *Ruminiclostridium* 9 spp. (rho = −0.40, *p* = 0.02), Ruminococcaceae UCG.005 (rho = −0.39, *p* = 0.03), Ruminococcaceae UCG.014 (rho = −0.39, *p* = 0.02) and *Ruminococcus* 1 spp. (rho = −0.47, *p* ≤ 0.01). Other correlations were identified and are shown in Figure 6.

### 3.7. Association of Breast Milk Microbiota with Variables

Through the mixed linear model, the demographic, biochemical, and clinical variables of the study were integrated with the most abundant ASVs. High specificity was identified, indicating that ASVs could be strictly associated with each variable. The employment relationship, birth weight, and gestational weight gain showed the highest number of associated ASVs. After adjusting the model (with the ASVs that showed a significant association (*p* < 0.05), the model was fitted to control the effects of other variables), a significant decrease in the *Peptococcus* spp. was found in vaginal delivery compared to C-section delivery; a significant increase in *Eubacterium* spp. was found in the overweight category during lactation compared to adequate BMI; *Aquabacterium* spp., *Acinetobacter*, *Lawsonella* spp., and *Chryseobacterium* spp. were found to be in a higher proportion in samples from women with inadequate gestational weight by deficiencies compared to adequacies (Table 4).

## 4. Discussion

The results of this research show that maternal nutritional status and the diet of women during gestation and lactation could modulate the microbiota of breast milk. Excessive gestational weight gain, low micronutrient intake, and a high intake of simple sugars and saturated fat could impact the content of bacterial genera that is of interest for infant health, while the consumption of micronutrients of interest, such as folates, could contribute to the presence of bacteria with probiotic potential.

Four dominant phyla were found in this sample in the order: Firmicutes, Actinobacteria, Proteobacteria, and Bacteroidetes. Investigations such as that of Urbaniak et al. [42] in breast milk from 39 Canadian Caucasian women and Togo et al. [43], in a systematic review, included a total of 15,849 samples from 38 countries and reported Proteobacteria and Firmicutes as dominant phyla in breast milk, while Actinobacteria and Bacteroidetes occurred in lower relative abundances.

A total of 644 bacterial genera were identified, which is higher than the genera reported by Zimmermann et al. [44] in a systematic review that included 44 studies and 2655 women from 20 countries, in which 590 genera were identified. The results at the genus level were consistent with those found by Padilhaet al. [8] in the milk samples from Brazilian women, who reported *Streptococcus*, *Staphylococcus*, and *Corynebacterium* as dominant genera. This was also reported by Kim SY et al. [45] in Korean women; specifically, *Staphylococcus* spp. and *Streptoccocus* spp. were reported as the dominant genera [46], which could suggest that regardless of the geographical location of the lactating woman, both genera are represented in this fluid and their colonization could be linked to the retrograde flow from the oral cavity of the infant [47]. Some *Streptococcus* spp. and *Staphylococcus* spp. species have been associated with infant health by preventing the colonization of pathogens such as *Staphylococcus aureus*, a risk factor for sepsis in newborns, through mechanisms including the release of peptides with antimicrobial properties and hydrogen peroxide production [48,49], which is of interest to the promotion of breastfeeding in all areas, including the clinical setting.

The genera *Lactobacillus* spp. and *Bifidobacterium* spp. are important for infant health. In this study, *Lactobacillus* spp. presented a relative abundance of 2.4%, which is higher than that reported in breast milk samples from Brazilian women (0.06%) [8] and lower than that reported in European (3.2%) [7] and Canadian women (3%) [42]. The presence of *Lactobacillus* spp. in breast milk is important for its probiotic potential in relation to health from the first years of life [50]. Breastfed infants, unlike those who receive infant milk formula, present a microbiota richer in *Lactobacillus* spp. and bifidobacterial; however, in this study, the abundance of bifidobacteria was low in agreement with that documented in other works (1.4%) [7].

Several factors are involved in the modulation of the breast milk microbiota. The relationship between microbiota and type of delivery is controversial since some studies have not reported significant differences [10,42,51] while other reports have; one example is Khodayar et al. [52], who identified higher abundances of total bacteria in the colostrum and transitional milk of women who had a cesarean delivery, and Cortés et al. [7] who determined a higher microbial richness in the milk of women who had undergone a cesarean delivery, producing results consistent with those found in the present study.

Some hypotheses about the origin of the microbiota of breast milk have been documented: the retrograde translocation of bacteria from the oral cavity of the infant, the mother’s skin, the use of breast pumps, the oro-mammary route, and the entero-mammary route, the latter explaining how some bacteria present in the maternal gut and how their metabolites could reach the mammary gland during late pregnancy and lactation through a process mediated by immune cells [47,53], thus shaping the breast milk microbiome. This provides a transient microbiota in the infant with great influence on the maturation of the immune system in extrauterine life [3]. Therefore, the intestinal microbiome of women during pregnancy and lactation could modulate the microbiota of human milk, suggesting the importance of an adequate diet and nutritional status of the mother to achieve a healthy microbiota that can subsequently colonize the breast milk that the infant receives.

In relation to maternal nutritional status, weight gain during gestation is important to ensure fetal growth and development. Weight gains that exceed or fall below the established recommendations have been associated with perinatal complications. In this study, it was observed that breast milk samples from women with gestational weight gain above the IOM recommendations [15] showed a tendency to lower alpha diversity: a finding that coincided with that reported by Cabrera et al. [54], who identified that women with an obese BMI during lactation and excessive weight gain during gestation tended to have a less diverse bacterial community in their milk, with a higher relative abundance of *Staphylococcus* spp., and lower relative abundance of *Bifidobacterium* spp. Contrary to what was reported by Lundgren et al. [10], when analyzing 155 breast milk samples, they observed greater alpha diversity in the milk microbiota of women with higher gestational weight gain. Other investigations have reported no differences in relation to maternal weight [55]. Our findings highlight the importance of nutritional surveillance for the control of gestational weight gain during pregnancy, not only because of its multiple implications for the health of the mother-child binomial [56] but also because of the impact it could have on the microbiota and early colonization of the infant.

According to the results obtained from our study, it was found that the consumption of simple carbohydrates during gestation and lactation was positively correlated with *Enterobacter* spp., while the intake of dietary fiber during the first trimester of lactation presented a negative correlation, which is important since this bacterial genus is characterized by opportunistic pathogenic species, which are associated with nosocomial diseases and hospital infections [57]. On the other hand, the intake of simple carbohydrates during lactation was negatively correlated with *Bifidobacterium* spp., an important bacterial genus for infant health, which is considered a primary colonizer of the gastrointestinal tract of the infant due to its ability to take advantage of the oligosaccharides in breast milk; its reduction has been associated with the development of metabolic diseases [58].

The above is relevant, finding that 72% of lactating women participating in this study exceeded the %AMDR [26] established in this country for simple carbohydrates, which was represented mainly by the intake of “panela”(raw sugar cane cubes, which is a typical product of some Colombian regions), sugar and sugary drinks. On the contrary, it is noteworthy that the intake of fiber was very low, and 97% of women did not achieve the recommendations.

Other relationships of interest have identified the consumption of folic acid during lactation was positively correlated with *Akkermansia* spp.: a genus that is considered a potential new generation probiotic with biotherapeutic actions for health in different metabolic disorders and other health alterations associated with intestinal dysbiosis [59]. In this study, it was identified that the main food sources of folate consumed by lactating women were cereals and fortified flours. In addition, some women reported continuing the consumption of their folic acid supplementation; however, 87% presented a risk of deficiency in the usual intake of this nutrient. Therefore, the consumption of food sources of this micronutrient, including legumes and green leafy vegetables, as well as folic acid supplementation in women at risk of deficiency, could be a dietary intervention strategy that favors the presence of probiotic bacteria such as *Akkermansia* spp. in breast milk.

In this group of lactating women, a high prevalence in risk of deficiency in the usual intake of micronutrients such as vitamin A, C, B6, B1, B3, zinc, and magnesium were identified, and it was also found that the consumption of foods belonging to fats, fruits and vegetables, meats, eggs, legumes, nuts, and seeds did not reach the recommendation proposed in the Dietary Guidelines for lactating women in Colombia [60]. This could negatively impact the microbial configuration of the mother and, therefore, that of the newborn. In Colombia, health programs have focused on nutrition during pregnancy, but the nutrition of lactating women has not been attended to, which has serious repercussions on the characteristics of human milk, its richness, and bacterial diversity, which is essential for the newborn.

Although this group of women enjoyed food and nutrition security at home, they presented an inadequate intake of macro and micronutrients, which may be conditioned by food choices that do not contribute to a diverse, healthy diet and favor the consumption of risk nutrients such as simple carbohydrates. The results of this and other research make it relevant to focus on lactating women because of the implications that their nutrition has on the milk microbiota and maturation of the immune system in the first years of life.

## 5. Limitations of the Study

During gestation, the use of the frequency of food consumption generated an overestimation of calcium intake. This is the first observational study conducted in this country, and future trials and intervention studies are needed to validate our findings.

## 6. Conclusions

This study is the first in Colombia to explore the relationship between nutritional status and feeding during gestation and lactation with the composition of breast milk microbiota. In this group, we observed that the diet of women could be related to genera of interest for maternal and child health; we observed a negative correlation between the lactation intake of simple carbohydrates and pregnancy intake with saturated fat and the genus *Bifidobacterium* spp.; furthermore, a positive correlation was identified between the lactation intake of folic acid and *Akkermansia* spp. These results contribute to new knowledge in maternal and infant nutrition and favor the bacterial ecosystem through interventions that contribute to healthy food choices and the feeding patterns of women during the reproductive cycle.

## Figures and Tables

**Figure 1 microorganisms-11-01812-f001:**
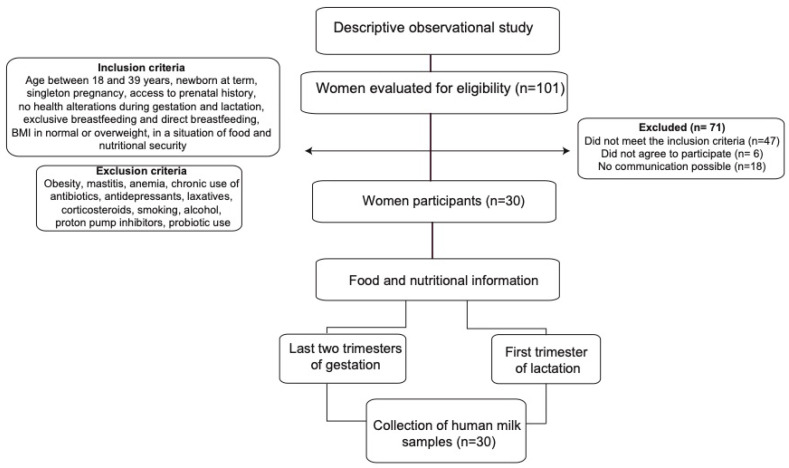
Flow diagram summarizing the identification and selection of participants and the methodology process.

**Figure 2 microorganisms-11-01812-f002:**
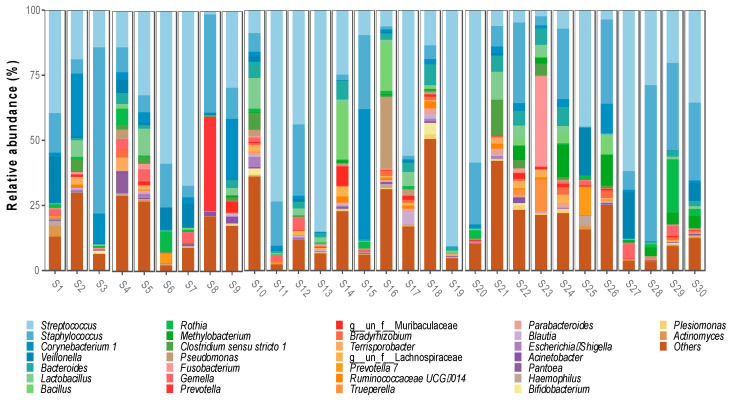
Bacterial genera identified in breast milk microbiota.

**Figure 3 microorganisms-11-01812-f003:**
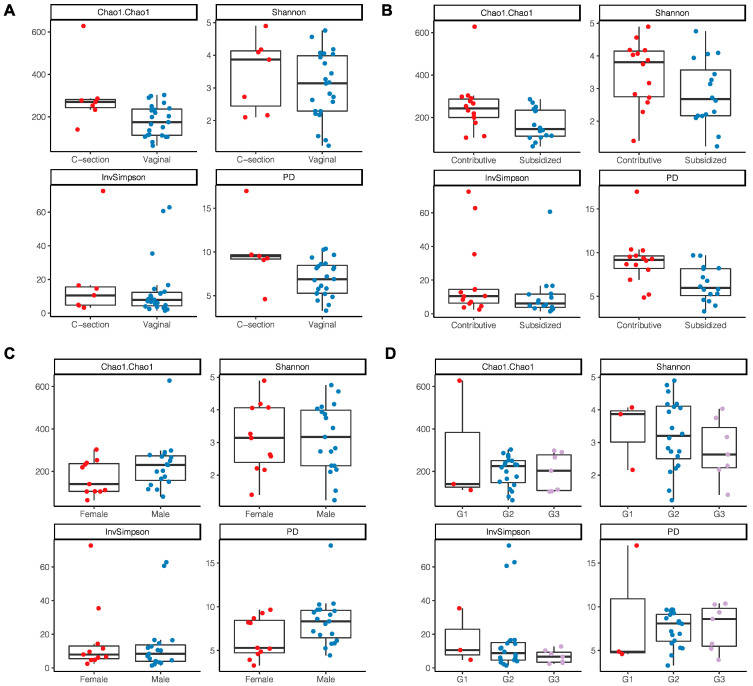
Alpha diversity indices of breast milk microbiota. (**A**) Type of delivery; (**B**) Employment relationship; (**C**) Baby’s gender; (**D**) Birth weight where G1: birth weight of 2700 g and <3000 g; G2: 3000 g and 3500 g; G3: >3500 g.

**Figure 4 microorganisms-11-01812-f004:**
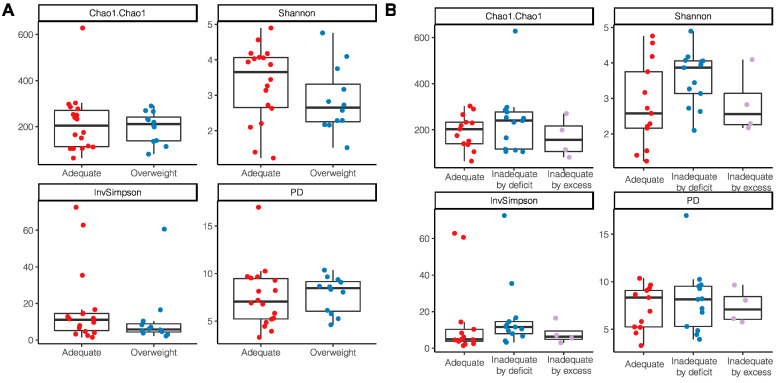
Alpha diversity indices of breast milk microbiota in relation to anthropometric indicators of nutritional status. (**A**) BMI of breastfeeding woman; (**B**) Gestational weight gain.

**Figure 5 microorganisms-11-01812-f005:**
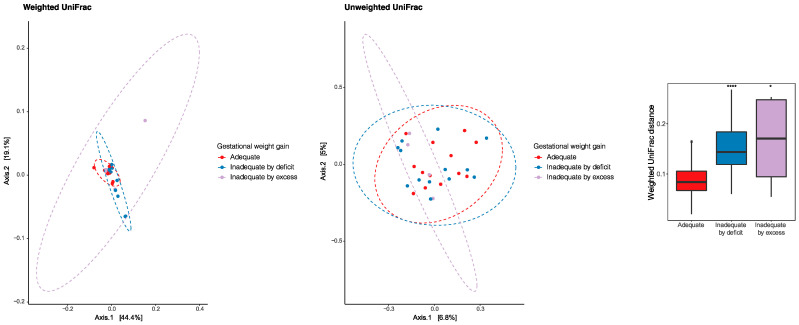
Principal coordinate analysis (PCoA) to identify a clustering pattern of microbial composition based on weighted and unweighted UniFrac distances and according to gestational weight gain. Pairwise beta diversity dissimilarities assessed by Unifrac distances and represented by a boxplot (* *p* < 0.05, **** *p* < 0.0001, Reference group = Adequate).

**Figure 6 microorganisms-11-01812-f006:**
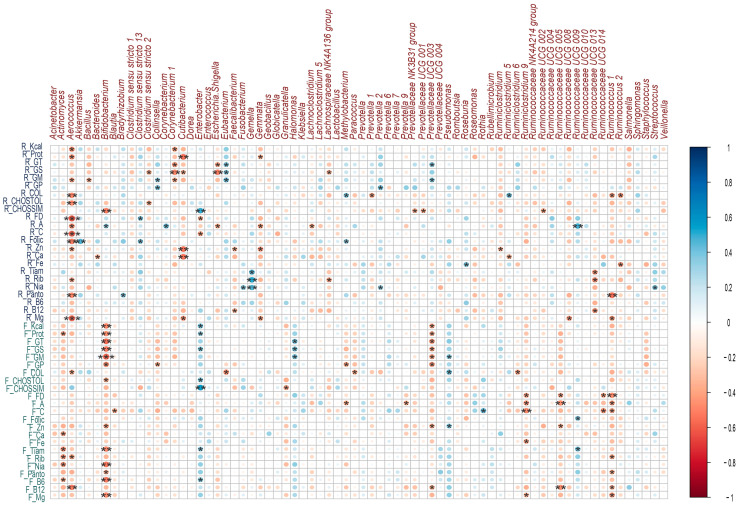
Correlations between breast milk microbial genera and nutrient intake during lactation and gestation. Heatmaps of Spearman rank correlations. Significant correlations (* *p* < 0.05, ** *p* ≤ 0.01, *** *p* ≤ 0.001). Blue circles represent positive correlations, whereas red circles show negative correlations. Note: the letter F and R on the left side of the heat map refers to Food Consumption Frequency and 24 h recall, respectively.

**Table 1 microorganisms-11-01812-t001:** Sociodemographic, gestational, and anthropometric characteristics of the breastfeeding women.

Variable	*n*	%
**Sociodemographics**		
**Marital Status**		
With partner	22	73.3
Without partner	8	26.7
**Level of schooling ***		
Elementary school	1	3.3
Secondary School	7	23.3
High school	11	36.7
Higher education	11	36.7
**Health Regime ***		
Contributory	14	46.7
Subsidized	16	53.3
None	10	33.3
**Zone of residence**		
Urban	13	43.3
Rural	17	56.7
**Gestational**		
**Pregnancy planning**		
Yes	12	40.0
No	18	60.0
**Previous pregnancies**		
No pregnancies	13	43.3
1–2	16	53.3
≥3	1	3.3
**No. of prenatal checkups**		
≤5	6	20.0
6–8	17	56.7
≥9	7	23.3
**Type of delivery**		
Vaginal	23	76.7
Cesarean section	7	23.3
**Anthropometric**		
**Pregestational BMI (Kg/m^2^)**		
Adequate	21	70.0
Thin	1	3.3
Overweight	8	26.7
**Breastfeeding BMI (Kg/m^2^)**		
Adequate	18	60.0
Overweight	12	40.0
**Gestational weight gain**		
Adequate	13	43.3
Deficit	13	43.3
Excess	4	13.3

Data presented as *n* (%). * Educational level and health status were classified according to national guidelines.

**Table 2 microorganisms-11-01812-t002:** Distribution and adequacy of energy intake, the percentage of individuals with intakes above and below the Acceptable Macronutrient Distribution Range (% AMDR) and prevalence of risk of deficiency for the usual intake of protein, vitamins and minerals.

Nutrients	< % Reference Value * % (SD)	> % Reference Value * % (SD)	Prevalence of Deficiency % (SD)	Adjusted Minimum	Adjusted Maximum	Adjusted Percentiles					Adjusted Mean (SD) ***
						5	25	50	75	90	
Calories (Kcal)	42.8 (0.11)	16.0 (0.12)		890	3419	1528	1917	2186	2455	2695	2185 (399)
Protein (g)	50.3 (0.14)	0.2 (0.01)	98.5 (0.03)	33.3	131.3	50.0	65.0	75.0	85.0	94.0	74.8 (15)
Total fat (g)	0.2 (0.01)	3.2 (0.12)		29.4	120.3	49.0	60.1	68.7	77.9	86.8	69.4 (13.2)
Saturated fat (g)	14.02(0.184)	86.0 (0.18)		9.28	50.30	18.80	24.40	28.40	32.70	36.60	28.60 (6.12)
Monounsaturated fat (g)				10.20	40.40	16.00	20.00	23.40	27.10	30.40	23.65 (4.97)
Polyunsaturated fat (g)				5.12	28.65	9.20	11.00	12.40	14.00	15.70	12.60 (2.32)
Cholesterol (mg)				70	1299	277	389	480	583	686	493 (145)
Total carbohydrates (g)	2.3 (0.10)	1.0 (0.06)		98.3	568.9	207.0	267.0	311.0	359.0	403.0	314.1 (68.0)
Simple carbohydrates (g)	28.0 (0.102)	72.0 (0.10)		1.9	243.2	18.0	48.0	77.0	109.0	144.0	82.8 (48.2)
Dietary Fiber (g)			3.1 (0.05) **	48.2	49.8	5.8	9.2	12.6	17.1	22.4	13.9 (6.6)
Vitamin A (ER)			49.8 (0.12)	172	13,063	371	599	903	1489	2638	1396 (1803)
Vitamin C (mg)			64.9 (0.10)	14	381	25	49	77	122	185	98 (75)
Folates (ugEFD)			87.3 (0.10)	49	929	130	202	273	368	478	299 (136)
Zinc (mg)			52.3 (0.09)	4.22	24.63	5.90	8.00	9.80	11.90	14.10	10.16 (3.00)
Calcium (mg)			30.2 (0.15)	239	2892	548	761	940	1147	1360	971 (293)
Iron (mg)			9.1 (0.10)	4.5	61.7	8.9	12.3	16.1	22.0	30.1	18.6 (9.5)
Thiamine (mg)			57.0 (0.17)	0.22	5.26	0.77	0.97	1.15	1.36	1.59	1.19 (0.30)
Riboflavin (mg)			0.6 (0.04)	0.91	7.65	1.53	1.87	2.18	2.59	3.05	2.29 (0.59)
Niacin (mg)			51.6 (0.10)	3.7	24.2	7.6	10.3	12.8	15.8	18.5	13.2 (3.9)
Panthotenic acid (mg)			18.8 (0.10) **	2.19	11.97	3.40	4.50	5.50	6.60	7.80	5.67 (1.63)
Vitamin B6 (mg)			59.8 (0.11)	0.74	3.91	1.03	1.31	1.58	1.93	2.34	1.68 (0.51)
Vitamin B12 (ug)			7.3 (0.10)	0.78	93.56	2.20	3.90	6.90	14.30	32.30	16.18 (37.20)
Magnesium (mg)			53.3 (0.10)	83	492	163	217	260	305	349	263 (65)

Data are presented as percentages (%), standard deviation (SD), minimum and maximum values, adjusted percentiles and adjusted mean. * Reference values for calories 90–110%; protein 14–20%AMDR; total fat 20–35%AMDR; saturated fat 10%AMDR; total carbohydrates 50–65%AMDR; simple carbohydrates 10%AMDR. ** Low risk of deficiency for dietary fiber and pantothenic acid. *** SD: standard deviation.

**Table 3 microorganisms-11-01812-t003:** Distribution of energy and nutrient intake during pregnancy.

Nutrients	Minimum	Maximum	Percentiles					
			5	25	50	75	90	Median (MAD) *
Kcal (Kcal)	1495	5895	1582.1	2036.25	2553	3163.75	4408.6	2553 (597.5)
Protein (g)	56.2	171.5	59.715	80.825	93.8	131.425	160.2	93.8 (24.5)
Total Fat (g)	37.9	135.5	45.94	58.45	71.05	93.4	113.95	71.1 (17.1)
Saturated fat (g)	15.6	56.36	17.909	22.165	28.055	36.52	47.35	28.1 (6.95)
Monounsaturated fat (g)	13.28	57.97	17.508	22.105	28.965	40.542	46.027	28.9 (8.16)
Polyunsaturated fat (g)	4.91	23.34	6.369	9.185	11.115	15.063	20.469	11.1 (2.59)
Cholesterol (mg)	108	1698	189	432.25	544.5	886.25	1174.6	544.5 (229)
Total carbohydrates (g)	145.1	1024.8	200.025	279.975	378.45	451.975	678.6	378.5 (97)
Simple carbohydrates (g)	2.6	323.9	8.905	42.55	76.2	112.85	135.34	76.2 (36.1)
Dietary Fiber (g)	11.2	63.9	12.8	20.075	23.8	37.6	42.46	23.8 (8.1)
Vitamin A (ER)	602	3761	668.7	1023.25	1394.5	1927.75	2252.8	1394.5 (443.5)
Vitamin C (mg)	109	840	124.45	222.25	254.5	437	658.1	254.5 (70)
Folates (ugEFD)	1043	2541	1130.05	1243.75	1461.5	1903.25	2114.9	1461.5 (257)
Zn (mg)	6.9	34.8	7.445	9.2	12.4	18.025	21.12	12.4 (4.4)
Fe (mg)	34.3	146.2	41.965	76.675	80.35	88.3	99.42	80.4 (5.15)
Thiamine (mg)	0.99	4.48	1.038	1.373	1.995	3.07	3.741	1.9 (0.67)
Riboflavin (mg)	1.43	6.3	1.484	2.053	2.855	4.442	5.476	2.9 (0.99)
Niacin (mg)	10.4	45.7	12.615	16.25	20.35	27.725	33.44	20.4 (5.75)
B6 (mg)	1.4	7.5	1.445	1.825	2.6	3.2	4.31	2.6 (0.65)
B12 (ug)	1.96	20.37	2.839	4.765	6.58	10.387	14.238	6.6 (2.33)
Mg (mg)	181	820	192.6	263.75	346	511	619.7	346 (97.5)

Data are presented as minimum, maximum, percentiles and the absolute deviation from the median. * MAD: median absolute deviation.

**Table 4 microorganisms-11-01812-t004:** Interaction between the most frequent ASVs (*p* < 0.05) and analyzed variables.

Variable	Groups ^1^	ASVS Associated	Top ASVs Associated	*p*-Value	Adjusted *p*-Value
Delivery mode	**C-Section**/Vaginal	2	*Bacillus*	0.0036	0.8707 (−0.21)
			*Peptococcus*	0.0041	**6 × 10^−4^** (−1.09 for Vaginal)
Employment relationship	**Contributive**/Subsidized	10	*Actinobacillus*	0.026612	**0.0190** (−2.03)
			*Mogibacterium*	0.030953	**0.0086** (−1.18)
			*Prevotellaceae NK3B31 group*	0.033343	**0.0136** (−2.00)
			*Corynebacterium*	0.038361	0.1172 (−1.20)
			*Gemella*	0.039113	0.3832 (−0.93)
			*Prevotella2*	0.040020	**0.0125** (−2.19)
			*Micrococcus*	0.040481	0.1062 (−1.57)
			*Kocuria*	0.046419	**0.007** (−2.08)
			*Phascolarctobacterium*	0.048834	**0.0495** (−1.56)
			*Ruminococcus1*	0.050555	**0.036** (−1.69)
Baby’s gender	**Female**/Male	3	*Acinetobacter* *Staphylococcus* *Alloprevotella*	0.003059 0.028668 0.047407	**0.014** (−1.83) 0.0962 (−0.95) 0.0882 (1.37)
Birth weight	**G1**/G2,G3	**47**	*Brachybacterium*	0.000319	**<0.001** (−0.98)
			*Clostridium sensu stricto 5*	0.001973	**<0.005** (−3.14)
BMI of the breastfeeding woman	**Adequate**/Overweight	3	*Eubacterium* *Aquabacterium* *Acinetobacter*	0.027936 0.040497 0.049149	**0.0254** (0.70 for Overweight) 0.0697 (−1.63) 0.0813
					(−1.31)
Gestational weight gain	**Adequate**/Inadequate by deficiencies, Inadequate by excess	5	*Aquabacterium* *Acinetobacter* *Bradyrhizobium* *Lawsonella* *Chryseobacterium*	0.00180 0.01088 0.03515 0.03831 0.04149	**0.008**, 0.7101 (2.99 for Inadequate by deficit, 0.43 for Inadequate by excess) **0.0090**, 0.3281 (2.03, 1.04) 0.0965, 0.0633 (0.93, 1.53) **0.0164**, 0.2976 (2.34, −1.41) **0.0267**, 0.8725 (1.86, 0.18)

^1^ Linear mixed model and adjustment by interactions. The reference group for the comparison is highlighted in bold. The most representative ASVs associated with each variable included in the model are shown; after adjusting for all factors, the last column shows the *p*-value and in parentheses the sense of the interaction.

## Data Availability

Not applicable.

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
