# Peer review of "Maternal Diet May Modulate Breast Milk Microbiota—A Case Study in a Group of Colombian Women"

_microorganisms, 2023, doi:10.3390/microorganisms11071812_

Round 1

Reviewer 1 Report

This study aimed to infer the causal relationship between maternal nutritional status & nutrient intake (two 24h-food recall/ one FFQ) and sociodemographic data with microbiome characteristic profile [16S ribosomal RNA (rRNA) sequencing, Illumina platform] in an epidemiological study (descriptive, analytical cross-sectional) involving a group of lactating women from Colombia (25 ± 6y, first-time mothers 43%, 60% normal BMI, n=30). Although the experimental design, execution, and analysis (descriptive/inductive) are robust enough, certain changes (narrative, format) could improve the study's uniqueness and scientific contribution. 

General. A) It is recommended that the latest version of the manuscript be reviewed by a native English speaker. B) Double-check the use of abbreviations throughout the manuscript and that their meaning has been included the first time they are cited.

Title. Although accurate, it does not reflect the whole picture: just diet? Are there not concurrent events that modify the microbiota in a synergistic or antagonistic way?.

Abstract. OK.

Introduction. The narrative of this section is somewhat scattered, due to the structure in small, non-interconnected paragraphs: A) It can be shorter and more concrete and it is recommended to build all  paragraphs effectively (see: https://doi.org/10.1111/1365-2435.13391, https://www.nature.com/scitable/topicpage/effective-writing-13815989/). B) It seems that important backup references were not included (nor in the discussion section; https://doi.org/10.3389/fmicb.2017.02100, 10.5223/pghn.2022.25.3.194 , https://doi.org/10.2217/fmb-2018-0317) including some that highlight the need of this new study (https://doi.org/10.21203/rs.3.rs-1739630/v2 , https://doi.org/10.15446/rev.fac.cienc.v12n1.99209).

Materials & Methods. OK.

Results. A) It is recommended to review the format of tables and figures (including their footnotes) according to the instructions for authors or based on similar articles published in this journal (e.g. https://doi.org/10.3390/microorganisms11041090). B) All figures should be provided with enough resolution (> 300 dpi)

Discussion. A) Is there any worth mentioning peculiarity related to the diet and living conditions of lactating Colombian women vs. others globally? B) The reported relative abundance [Streptococcus spp.> Staphylococcus spp.> Corynebacterium spp.> Veillonella spp.) is a common feature in breast milk? or is it a specific firm for this group of women? C) It is recommended, based on narrative and systematic reviews on the subject, to generalize the similarities or differences of this study with others of a similar nature.

References. The ratio of new/old references is correct. However, several references are incorrectly formatted according to the guidelines of this journal (https://www.mdpi.com/journal/microorganisms/instructions).

The manuscript´s grammar and syntax should be reviewed

Reviewer 2 Report

The bacterial microbiome has received considerable attention in many species, including humans, over the past 15 years.  How the microbiome is established is an important question for neonatal health and such studies as vaginal birth vs c-section and the role of maternal and family and environmental transfer early in life are all important. 

            My first major comment is that the present study is very focused on the role of diet in regulating the microbiome of breast milk and it does not address what effect this microbiome has  on the infant and we shall briefly return to this question since the neonatal effect may be ultimately what we are interested in. 

One question which should be addressed in the Discussion is how the

Could the intestinal bacterial microbiome effect the breast milk microbiome.   Bacteria can traverse from the intestine to other parts of the body and not only the intestinal but oral microbiome are likely potential candidates for ‘direct’ alteration by population of milk microbiome.   Secondly the intestinal microbiome may make metabolites which are found in the breast milk and may effect the breast milk microbiome.  That diet might yield components, digested or undigested, that may end up in breast milk and effect the microbiome is a hypothesis presented well in the manuscript. 

            As my second major coment, an important facet of the studies not presented is the effect of different breast milk microbiota on the infant.  This question is important and large, but one facet easily accomplished within the repertoire of technologies used in the present studies is to genotype the neonatal stool microbiome.   If this data is available, then this would greatly enhance the study.  Of note, the authors have discussed some of this topic and for all of their conclusions, they acknowledge limitations in the data and have been conservative in interpretation.   With greater, deeper mechanistic investigations, these questions shall be answered.  The present study is a good start and merits publication.   

Reviewer 3 Report

Title: I'm sorry but it is already known that the mother's diet modifies the microflora of the breast milk. So please consider changing the title of the article as follows: Maternal Diet May Modulated Breast Milk Microbiota. A case study in a group of Colombian Women. Or to something else. It’s up to you.

L 50-52: Please put references of the last three years or if there are none, highlight the importance of the specific references.

L 62-63: “In relation to nutritional status, ……….in relation to Body Mass Index (BMI) and gestational weight gain”. Please rephrased.

The introduction needs enrichment and data addition.

L 77-80: “The objective of this study was to analyze the effects of food and nutritional status during gestation and the first trimester of lactation on the microbiota of breast milk in a group of healthy lactating women in Colombia”. Several studies have been written on this topic. Why the study of Colombian women is something more special? Please explain it.

Analysis of microbiota data: Were sequences trimmed or excluded as expected errors? Accessions numbers for sequencing and metagenomics analyses.

The figures have a low resolution in the version available.

L 466-468: “The results of this research show that the maternal nutritional status and the diet of the woman during gestation and lactation could modulate the microbiota of breast milk, through the entero-mammary pathway[42].” Which study? Yours? The reference (42) you have placed proves this conclusion.  So why is needed your study?

The discussion is weak. Please enriches the section.

Round 2

Reviewer 3 Report

I have no other comment. Congratulations on your work.